# Validation of a New HPLC-DAD Method to Quantify 3-Deoxyanthocyanidins Based on Carajurin in Medicinal Plants and for Chemical Ecology Studies

**DOI:** 10.3390/plants12051057

**Published:** 2023-02-27

**Authors:** Paulo Victor R. Souza, Vanessa G. F. C. Martins, Maria do Socorro S. Chagas, Carla J. Moragas-Tellis, Maria D. Behrens, Davyson L. Moreira

**Affiliations:** 1Laboratory of Natural Products for Public Health, Pharmaceutical Technology Institute (Farmanguinhos), Oswaldo Cruz Foundation, Rio de Janeiro 21041-250, Brazil; 2Environmental Toxicology Laboratory, National School of Public Health (ENSP), Oswaldo Cruz Foundation, Rio de Janeiro 21041-361, Brazil; 3Postgraduate Program in Translational Research in Drugs and Medicines, Pharmaceutical Technology Institute (Farmanguinhos), Oswaldo Cruz Foundation, Rio de Janeiro 21041-250, Brazil; 4Laboratory of Natural Products, Rio de Janeiro Botanical Garden Research Institute, Rio de Janeiro 22460-030, Brazil

**Keywords:** anthocyanidins, color, carajurin, medicinal plants, HPLC, development, validation, *Arrabidaea chica*

## Abstract

Anthocyanidins are found in nature mainly as pelargonidin, cyanidin, peonidin, delphinidin, petunidin, and malvidin derivatives. These compounds are found free or as glycoside derivatives which are responsible for the color (red, blue, and violet) of some foods and are responsible for attracting seed dispersers. They are grouped into 3-hydroxyanthocyanidins, 3-deoxyanthocyanidins (3D-anth), and *O*-methylated anthocyanidins. A new method was developed and validated to quantify 3D-anth in plant-rich extracts. To test the new method, *Arrabidaea chica* Verlot was selected as it is widely used in folk medicine, and it is rich in 3D-anth. The new method was developed by HPLC-DAD and expressed 3D-anth as carajurin content. Carajurin was chosen as the reference standard due to its role as a biological marker for the antileishmanial activity for *A. chica*. The selected method used a silica-based phenyl column, a mobile phase composed of potassium dihydrogen phosphate buffer, acetonitrile, and methanol, in a gradient elution mode and detection at 480 nm. The method reliability was confirmed by verifying selectivity, linearity, precision, recovery, and robustness. This method contributes to quality control and development of a possible active pharmaceutical ingredient from *A. chica* as well as it can be used to evaluate 3D-anth in plant extracts with chemical ecology interest.

## 1. Introduction

Anthocyanidins feature a flavilium cation in their structure. Six of 19 different types of anthocyanidins (pelargonidin, cyanidin, peonidin, delphinidin, petunidin, and malvidin) prevail in nature (Figure 1) [1]. These compounds are found in glycoside forms (anthocyanins) and are responsible for the color of some foods, especially fruits such as cranberry, blueberry, grape, sweet cherry, strawberry, marionberry, blackberry, raspberry, apple, peach, nectarine, and plum [2]. In plants, anthocyanidins offer protection from UV, and they are active against microorganisms, including virus, bacteria, and fungus. Besides, anthocyanidins may attract seed dispersers, an important ecological function. They are grouped into 3-hydroxyanthocyanidins, 3-deoxyanthocyanidins (3D-anth), and *O*-methylated anthocyanidins [3,4]. The 3D-anth differs from the other by the absence of a hydroxyl at position 3 of the *C*-ring. Considered the anthocyanins chemical precursors, they are found in flowers and fruits and responsible for their impressive blue and purple colors. Occurring in a limited number of plant species, including the medicinal *Arrabidaea chica* (Humb. & Bonpl.) B. Verl., they show a redder color than the corresponding anthocyanins [5]. To date, the literature described the extraction, isolation, and chemical characterization of four 3D-anth from *A. chica:* 6,7,3′,4′-tetrahydroxy-5-methoxy-flavylium (3′-hydroxy-carajurone) (1), 6,7,4′-trihydroxy-5-methoxy-flavylium (carajurone) (2), 6,7,3′-trihydroxy-5,4′-methoxy-flavylium (3′-hydroxy-carajurin) (3) and 6,7-dihydroxy-5,4′-dimethoxy-flavylium (carajurin) (4) [6,7]. Their chemical structures are shown in Figure 1.

Despite having the same biosynthetic origin as other flavonoids, anthocyanidins have specific structural characteristics. Due to a continuous delocalization of the π electrons along the A, B and C rings of their backbone, the absorption band of anthocyanins occurs in the visible range, varying between 480 and 540 nm. In the specific case of 3D-anth, the highest absorption occurs at 480 nm [8,9,10]. From these data, a first screening was performed in a full-scan experiment (200–600 nm). As results, chromatograms at 480 nm show the maximum wavelength (λ_max_) for the 3D-anth, confirming the literature data; therefore, this wavelength was adopted in the monitoring of these class of substances.

*Arrabidaea chica* was chosen for these experiments because it is in the National List of Medicinal Plants of Interest (Renisus) to the Unified Health System (SUS) by the Brazilian Ministry of Health (MH) in 2009. This list includes 71 plant species being used in treatments or whose pharmacological activities have been proven in studies. According to the MH, these species have the potential to reach production and benefit the SUS, including *A. chica* [11]. Native to the Amazon region, *A. chica*, popularly named crajiru and pariri, belongs to the Bignoniaceae family. It is also found in American and African tropical regions [6,12]. Its leaves are traditionally used as teas to treat skin inflammation, gastrointestinal disorders, mycoses, wounds, ulcers, and other ailments [13,14,15].

Carajurin is the most studied 3D-anth from *A. chica*. Its isolation and chemical characterization are reported by several research groups [6,7,16]. Recently, carajurin was proposed as biological marker for the antileishmanial activity of hydroalcoholic extracts of *A. chica*, as activity levels against the promastigote forms of *Leishmania amazonensis* correlate with the carajurin content [16]. Subsequently, a bioguided study confirmed, when comparing activities of the crude hydroalcoholic extract from *A. chica*, an anthocyanidins-rich fraction, and its isolated anthocyanidins, that carajurin has the highest activity of all examined options [17]. Then, a set of experimental evidence and literature reports consolidated the proposition of carajurin as a biological marker of *A. chica* for antileishmanial activity [18]. Lately, it showed lethal effect against promastigote forms of *L. amazonensis*, inducing ultrastructural changes, mitochondrial membrane potential decrease, and increased ROS production, which together induced cell death by late apoptosis [19].

Carajurin might be exclusive to the genus *Arrabidaea* and possibly limited to the species *A. chica* due to lack of mention of its isolation from other plant species [16,20]. But with new and fast validated methods for analysis of 3D-anth, other plant material can be monitored in a medicinal and chemical ecological point of views. Besides, despite long use and its Renisus-inclusion, no official compendium still regulates the quality control of *A. chica*. Quality is challenging when handling plant species, especially considering industry’s aims of obtaining safe and effective pharmacological products. The World Health Organization (WHO) recommends methods to assess the quality of medicinal plant materials [21]. The complexity of extracts’ chemical composition and their potential seasonal and regional variation requires plant materials’ standardization. The WHO accepts chromatographic profile analyses for identifying and evaluating the quality of medicinal plants and the variability of the chemical components of extracts [22,23]. The standardization of extracts is essential for herbal medicines production, enabling the administration of known concentrations of compounds and better pharmacological responses [24]. This is a key step for their use as active pharmaceutical ingredients (API).

Considering chemical ecology studies, analytical methods capable of performing fast and robust analyzes can contribute to studies, for example, of substances that are responsible for the colors of the plants, including anthocyanidins. These studies are extremely important to verify if a seed disperser is also attracted by the color, consequently, by the anthocyanidin and anthocyanin content [3,4].

To date, only three papers show analytical methods by High-Performance Liquid Chromatography (HPLC) for quantification of anthocyanidins from *A. chica* [6,25,26]. However, they lack reproducibility, good resolution for the carajurin peak, and require long analysis times (30 to 45 min). None of them presented information about development and/or validation, i.e., they do not guarantee reliability.

## 2. Results and Discussion

### 2.1. Method Development

The analytical conditions tested (Table 1) followed HPLC’s method development procedures [27,28,29]. The first attempts to develop a practical method for quantification of 3D-anth from *A. chica* used different proportions of acidified water (H_2_O)/acetonitrile (ACN)/methanol (MeOH) (*v*/*v*/*v*) in isocratic conditions. However, none of the five different tested proportions provided visualization of peaks for carajurin and other anthocyanidins present in the extract. Thus, the same solvent mixture was tested with a gradient elution, and visualized 3D-anth’s peaks, but being only registered at the end of each chromatogram. This led to longer analysis time (from 12 to 18 min). Aiming to improve resolution, asymmetry, and width of the peak base, further analytical conditions were changed: new mixtures of the eluents; chromatographic columns; flow of the mobile phase; and gradient programs; totaling 45 attempts. For each small change resulting in improvement of carajurin peak, the conditions were maintained; if not, the previous condition was returned, and tested a new condition, until obtaining the best analytical conditions (Table 2). Figure 2 shows the resulting chromatogram of carajurin in the new method. The carryover test was monitored intra-day and inter-day by blank injections. A small peak was detected but its area in mAU was closer to limit of quantification (LDQ) (Figure 3A).

### 2.2. Method Validation

After establishing the best chromatographic parameters, we proceeded with the analytical validation. The newly developed analytical method (Table 2) was validated for selectivity, linearity, intra-day and inter-day precision, accuracy, detection and quantification limits, recovery, and robustness [30,31].

#### 2.2.1. Selectivity

Aiming to verify the absence of interferences in the detection window of the carajurin peak, the sample diluent (ACN:MeOH, 75:25, *v*/*v*) and an extract containing carajurin were evaluated. The chromatogram of the diluent (Figure 3A) shows no peaks, while in the one for the *A. chica* hydroalcoholic extract (ACHE) (Figure 3B), there is no interference close to the Rt of carajurin, which demonstrates the selectivity of the method.

#### 2.2.2. Linearity

Linearity was evaluated by analytical curves from carajurin diluted solutions (1; 2; 3; 4; 5, and 6 µg/mL). Analyses were in triplicate on three different days. No outliners were registered, and the average linear correlation coefficient (*r*^2^) was 0.9970, higher than the minimum value (0.99) described by the adopted references [30,31], demonstrating the method’s linearity. The linear regression equation used for quantification of samples was ABS = (104,771.3 ± 1873) × Concentration − (58,184 ± 9974). The points distribution in the analytical curve showed to be homoscedastic.

#### 2.2.3. Intra-Day and Inter-Day Precision

In the evaluation of precision, three concentrations of carajurin standard (1, 3, and 6 µg/mL) were used in triplicate at three different times on the same day (intra-day precision) and three different days (inter-day precision). The triplicate means, the standard deviation (SD), and the relative standard deviation (RSD) of analysis were calculated (Table 3 and Table 4). No significant differences were observed in results (*p* > 0.05). Additionally, considering that the variation values were lower than the established limits of RSD (<15%) [30,31], the method is precise.

#### 2.2.4. Detection and Quantification Limits

The limits of detection (LOD) and quantification (LOQ) were evaluated through the relation between signal and noise from the baseline in the chromatogram for successive dilutions of the carajurin stock solution. According to the reference guides, the ideal signal-to-noise ratios are 1:3 for LOD and 1:10 for LOQ. Considering these parameters, LOD was determined at 125 ng/mL and LOQ at 200 ng/mL (LOQ), therefore, compatible with a DAD-UV-Vis detector.

#### 2.2.5. Accuracy

The accuracy of the method was evaluated as percentage comparison between the theoretical concentrations (analytical curve points) and the ones experimentally obtained. As the variations found are within the limits allowed by the validation standards (85–115%), the method can be considered accurate (Table 5).

#### 2.2.6. Robustness

Four parameters and their variations were separately analyzed to evaluate robustness, meaning the entire method remained following nominal conditions, except for the variation of the parameter currently evaluated. This influence was evaluated at the lowest and highest concentration of the analytical curve. Acceptable variation is equal or bellow RSD of 15%. The highest variation was just over 2% and the method can be considered robust (Table 6).

#### 2.2.7. Recovery

This parameter was evaluated using carajurin-spiked samples at the highest and lowest concentrations (1 and 6 µg/mL) and the percentage comparison between the standard and the enriched solutions. Table 7 shows the results. Recovery rates were within the specification range determined by the legal references used (85–115%).

### 2.3. Comparison of the Newly Developed and Validated Method with Those from Literature

Some HPLC methodologies address analyses of *A. chica* extracts [32,33,34,35,36,37,38]. However, few studies specify a methodology for quantification of anthocyanidins in this species (Table 8). Prior to developing our methodology, several attempts proved unsuccessful in reproducing them. Additionally, none of these studies described the development and validation of their methodologies. For example, those methods developed by Paula et al. [25,26], followed by Devia et al.’s methodology [6], differ from ours in column dimensions, flow rate, and total analysis time. Taffarello et al.’s HPLC-DAD method [39] lacked a complete specification of the elution gradient, which precludes the determination of the total analysis time, thus being unreproducible.

The method presented herein was compared to those from the literature. It shows improvements concerning the peak width of the carajurin (approximately 1.0 min in our method vs. about 3.0 min in Devia et al. [6]), and total analysis time (18.0 min in this study vs. 45.0 min in Paula et al. [25]). These enhancements mean better chromatographic resolution and significant time saved, enabling more analyses per time and, consequently, lower mobile phase consumption. Other parameters were calculated to carajurin standard, such as peak asymmetry (0.9), peak tailing factor (1.1) and UV spectrum purity (98%).

### 2.4. Quantification of 3-Deoxyanthocyanidins (3D-Anth) Expressed as Carajurin

We quantified the 3D-anth in ACHE using the new developed and validated analytical method. The objective was to test the quantification efficacy of the new method in a more complex matrix, in addition to the tests already carried out during the development and validation stages. Figure 3B shows the obtained chromatogram of ACHE. Four 3D-anth were assigned (1–4), of which we identified carajurin (3) by comparison with the isolated standard (Figure 2) at Rt 14.2 min and co-injection. The anthocyanidins content in the extract was quantified through the linear regression equation of the analytical curve = ABS = (104,771.3 ± 1873) × Concentration − (58,184 ± 9974) and values are shown in Table 9. A previous comparison between four morphotypes of *A. chica* in two seasonal collections [16], quantifying carajurin in all extracts, took 90 min, a much longer analysis time. Further seasonal analyses based on the herein presented method will be more cost-effective.

Once adjusted, this method may be used to quantify 3D-anth in other plant materials or 3D-anth-rich extracts, considering its relevant gain in resolution and analysis time.

## 3. Materials and Methods

### 3.1. Plant Material Collection and Identification

Leaves from *A. chica* were cultivated and collected at Farmanguinhos/Fiocruz Atlantic Forest Campus, municipality of Rio de Janeiro, State of Rio de Janeiro, Brazil (S22.9406° W43.4046°). Plant material was identified by Dr. Marcus Felipe Oliveira da Silva at the Botanical Collection of Medicinal Plants (CBPM) of Farmanguinhos/Fiocruz, where a voucher specimen is deposited and registered under the number CBPM668. The plant material was compared with those deposited at the Herbarium of the Botanical Garden of Rio de Janeiro (JBRJ) and CBPM. All studies with *A. chica* were registered at the Genetic Heritage Management Council under the code A670412.

### 3.2. Extract Preparation and Carajurin Isolation

One gram of ground dried leaves (knife mill after oven at 60 °C with forced air circulation) was extracted by maceration in ultrasonic bath (Quimis, Diadema, Brazil) at room temperature in 10 mL ethanol:water (7:3, *v*/*v*) for 30 min. The obtained ACHE was filtered in a 0.45 μm PTFE filter (Merck Millipore, Darmstadt, Germany) before analyzed in a High-Performance Liquid Chromatograph coupled to Diode-Array Detector (HPLC-DAD). Carajurin was isolated and characterized [16]. The determined purity was 98% by HPLC-DAD [16].

### 3.3. Reagents

The HPLC grade: methanol (CAS 67-56-1) and acetonitrile (CAS 75-05-8) (from Tedia, Fairfield, CT, USA); and analytical reagent grade: acetonitrile (for robustness test); ammonium acetate (CAS 631-61-8); potassium dihydrogen phosphate (CAS 7778-77-0); phosphoric acid 85% (CAS 7664-38-2) and acetic acid (CAS 64-19-7) (from Merck, Darmstadt, Germany). Ultrapure water (type 1 water—18.2 MΩ·cm)/conductivity 0.055 μS/cm at 25 °C) was obtained by a Milli-Q water system by Merck (Darmstadt, Germany).

### 3.4. Method Development (Equipment and Conditions)

Chromatographic analyses (HPLC-DAD) carried out at the Laboratory of Environmental Toxicology, Fiocruz, with a Shimadzu Nexera XR^®^ chromatograph and a Shimadzu UV-VIS detector with diode array SPDM20A (CBM20A controller, DGU20A degasser, LC20AD binary pump, CTO20A oven, and SILA20A auto-injector). Chromatograms were analyzed with a Shimadzu LabSolutions Software 5.92v (Shimadzu, Japan).

Parameters were tested to improve analytical conditions. Columns were silica-based C18 Ascentis (250 mm × 4.6 i.d.; 5 μm particle size) and a modified silica phenyl Ascentis (250 mm × 4.6 mm i.d.; 5 μm particle size), both from Supelco (Merck, Darmstadt, Germany). Mobile phase tests tried different pH values, combining ultrapure water (pH 3.0, with anhydrous acetic acid), acetonitrile, methanol, and potassium dihydrogen phosphate buffer and ammonium acetate buffer. All buffer solutions were filtered through a 0.45 μm PVDF filter (Merck-Millipore, Darmstadt, Germany) before use.

In addition, we examined mobile phase flow, isocratic or gradient elution mode, and total time analysis, in this validation. Keeping oven temperature at 50 °C, analyses were performed in triplicate (20 μL sample injection), and 3D-anth detected at 480 nm.

### 3.5. Method Validation

We followed the guidance for validation from the resolution RDC 166/2017 of the Brazilian Health Surveillance Agency–Anvisa [30] and DOQ-CGCRE-008 of the Brazilian National Institute of Metrology, Quality and Technology (Inmetro) [31]. Analytical parameters evaluated were selectivity, linearity, intra-day and inter-day precision, accuracy, detection and quantification limits, robustness, and recovery. Isolated carajurin was used as the reference standard. Analytical curves were obtained from carajurin standard solutions—stock solution (200 μg/mL) in acetonitrile:methanol (75:25, *v*/*v*); and at 1, 2, 3, 4, 5, and 6 μg/mL (final concentrations). All determinations were done with a 20 μL injection volume and detection at 480 nm. Selectivity was obtained from the analysis of sample diluent (acetonitrile:methanol, 75:25, *v*/*v*), carajurin solution, and *A. chica* carajurin-containing extract samples. Linearity was achieved from three calibration curves obtained on three different days. The evaluation of precision was with three concentrations (1, 3, and 6 µg/mL) in triplicate at three different times on the same day (intra-day precision) and in three different days (inter-day precision). Accuracy was evaluated as percentage comparison between theoretical concentrations (analytical curve points) and the experimentally obtained. Limits of detection (LOD) and quantification (LOQ) were established from serial dilutions and observation of the ratio signal/noise (s/n) (LOD: s/n ≥ 3; LOQ: s/n ≥ 10). Robustness tests were carried out based on the analysis of the influence of four analytical parameters (mobile phase flow rate; oven temperature; brand of acetonitrile; and detection wavelength) and their respective variations in two different concentrations (1 and 6 μg/mL). Recovery was obtained from carajurin-spiked samples at the highest and lowest concentrations (6 and 1 μg/mL, respectively), followed by determining the concentrations added.

### 3.6. Preparation of Extract Samples

A total of 1000 µL of acetonitrile:methanol (both HPLC grade, Tedia) (75:25, *v*/*v*) was added to 10 mg of each extract, previously weighed in a 4 mL vial. The vial was sealed and brought to ultrasonic bath for 10 min with occasional swirling. The sample was vortexed to mix thoroughly and filtered (0.45 µm PTFE filter; Merck, Darmstadt, Germany) before further analyses.

### 3.7. Statistical Analysis

Results are noted as absorbance mean and standard deviation. Tukey’s test (ANOVA) was used to compare intra-day and inter-day precision (*p* < 0.05) using STATISTICA software version 10 (StartSoft Inc., Tulsa, OK, USA).

## 4. Conclusions

This paper presents a newly developed and validated method using HPLC-DAD to quantify 3D-anth expressed as carajurin. It showed, as a major advantage, a better peak definition of carajurin given by the peak asymmetry, tailing factor and peak width of this compound. In addition, the present method proves to be faster and more economical than those already existing in the literature, with a shorter total analysis time and in addition to the use of a silica-based phenyl column. Its validation demonstrated reliability by selectivity, linearity, precision, accuracy, recovery, and robustness, which demonstrates that the method is reproducible. Thus, the results will contribute to setting parameters for the quality control of medicinal plants rich in 3D-anth, such as *A. chica,* as well as anthocyanidins-rich extracts, to obtain active pharmaceutical ingredients or for chemical ecology studies.

## Figures and Tables

**Figure 1 plants-12-01057-f001:**
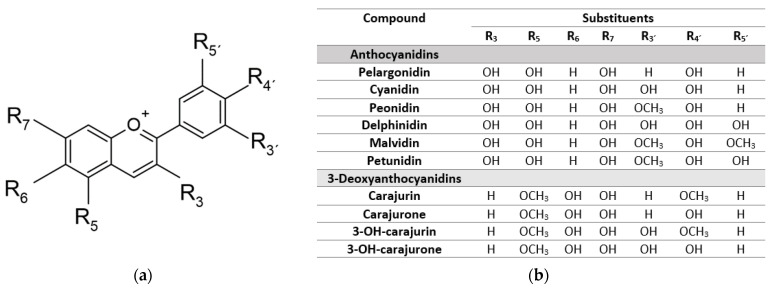
Chemical structure of anthocyanidins (**a**) for the six major types found in nature and the less common 3-deoxyanthocyanidins (3D-anth) (**b**).

**Figure 2 plants-12-01057-f002:**
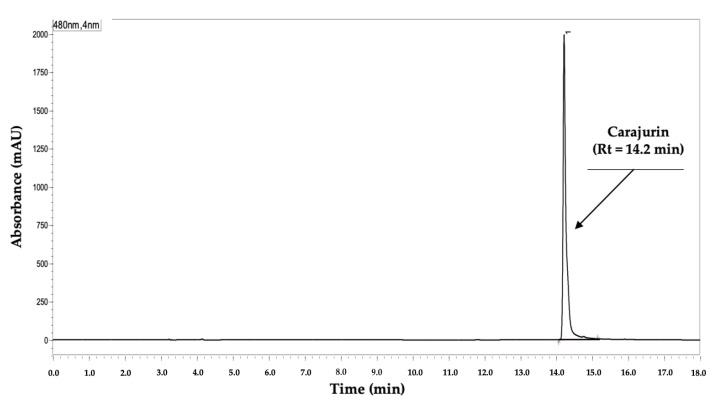
Chromatogram of carajurin solution (100 µg/mL) at 480 nm.

**Figure 3 plants-12-01057-f003:**
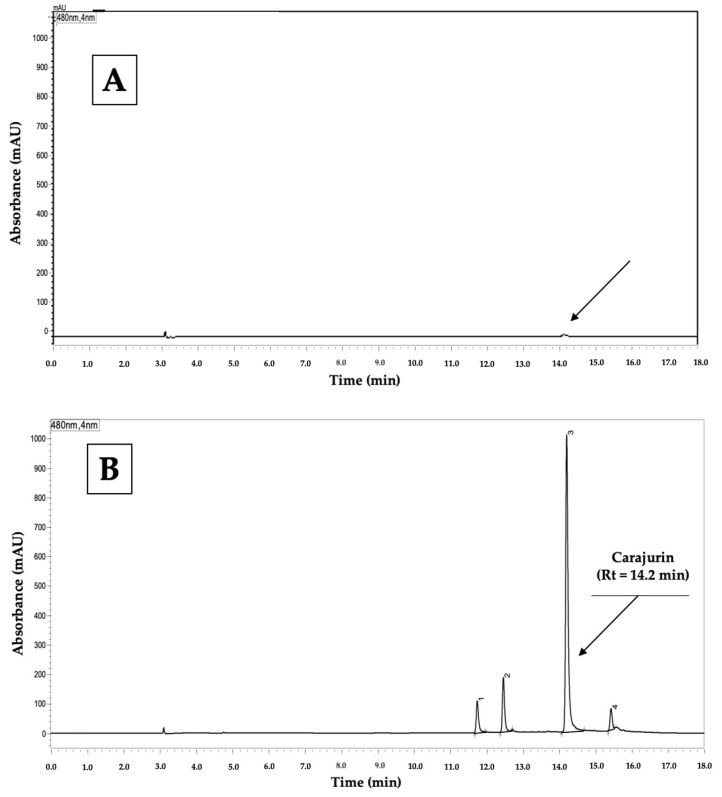
(**A**) Chromatogram for the analysis of the diluent (ACN:MeOH, 75:25, *v*/*v*) at 480 nm. A small peak observed in the carajurin chromatographic window is due to carryover, but it is closer the LDQ (black arrow). (**B**) Chromatogram for the analysis of the *Arrabidaea chica* hydroalcoholic extract (ACHE) at 480 nm.

**Table 1 plants-12-01057-t001:** Tested conditions for developing a new quantification method for carajurin.

Condition	Column	Mobile Phase A/B/C (*v*/*v*)	Gradient Program [Time (min)/%B/%C]	Total Analysis Time (min)	Flow Rate (mL/min)
1	C-18	H_2_O (pH 3)/ACN/MeOH (50:50:0)	Isocratic system	12	1.0
2	C-18	H_2_O (pH 3)/ACN/MeOH (30:70:0)	Isocratic system	12	1.0
3	C-18	H_2_O (pH 3)/ACN/MeOH (27:70:3)	Isocratic system	12	1.0
4	C-18	H_2_O (pH 3)/ACN/MeOH (15:80:5)	Isocratic system	12	1.0
5	C-18	H_2_O (pH 3)/ACN/MeOH (0:70:20)	Isocratic system	12	1.0
6	C-18	H_2_O (pH 3)/ACN/MeOH	0/2/8–3/12/8–9/95/0–9.1/2/8–11/2/8	11	1.0
7	C-18	H_2_O (pH 3)/ACN/MeOH	0/2/8–3/12/8–9/95/0–9.1/2/8–15/2/8	15	1.0
8	C-18	H_2_O (pH 3)/ACN/MeOH	0/2/8–3/12/8–9/100/0–9.1/2/8–15/2/8	15	1.0
9	C-18	H_2_O (pH 3)/ACN/MeOH	0/2/8–3/12/8–9/90/0–9.1/2/8–15/2/8	15	1.0
10	C-18	H_2_O (pH 3)/ACN/MeOH	0/2/8–3/12/8–9/88/0–9.1/2/8–15/2/8	15	1.0
11	C-18	H_2_O (pH 3)/ACN/MeOH	0/2/8–3/12/8–12/88/0–12.1/2/8–15/2/8	15	1.2
12	C-18	H_2_O (pH 3)/ACN/MeOH	0/2/8–3/12/8–12/90/0–12.1/2/8–15/2/8	15	1.2
13	C-18	H_2_O (pH 3)/ACN/MeOH	0/2/8–3/12/8–12/95/0–12.1/2/8–15/2/8	15	1.2
14	C-18	H_2_O (pH 3)/ACN/MeOH	0/2/8–3/12/8–12/85/0–15/2/8–18/2/8	18	1.2
15	C-18	H_2_O (pH 3)/ACN/MeOH	0/2/8–3/12/8–12/80/0–15/2/8–18/2/8	18	1.2
16	C-18	H_2_O (pH 3)/ACN/MeOH	0/2/8–3/12/8–12/50/0–15/2/8–18/2/8	18	1.2
17	C-18	H_2_O (pH 3)/ACN/MeOH	0/2/8–3/12/8–12/70/0–15/2/8–18/2/8	18	1.2
18	Phenyl	H_2_O (pH 3)/ACN/MeOH	0/2/8–3/12/8–9/80/0–12/2/8–18/2/8	18	1.0
19	Phenyl	H_2_O (pH 3)/ACN/MeOH	0/2/8–3/12/8–12/90/0–15/2/8–18/2/8	18	1.0
20	Phenyl	H_2_O (pH 3)/ACN/MeOH	0/2/8–3/12/8–9/80/10–12/2/8–18/2/8	18	1.0
21	Phenyl	H_2_O (pH 3)/ACN/MeOH	0/2/8–3/12/10–9/80/15–12/2/8–18/2/8	18	1.0
22	Phenyl	H_2_O (pH 3)/ACN/MeOH	0/2/8–3/12/8–9/65/15–12/2/8–18/2/8	18	1.0
23	Phenyl	H_2_O (pH 3)/ACN/MeOH	0/2/8–3/12/10–9/70/10–12/2/8–18/2/8	18	1.0
24	Phenyl	H_2_O (pH 3)/ACN/MeOH	0/2/8–3/12/10–9/65/10–12/2/8–18/2/8	18	1.0
25	Phenyl	H_2_O (pH 3)/ACN/MeOH	0/2/8–3/12/10–9/60/10–12/2/8–18/2/8	18	1.0
26	Phenyl	H_2_O (pH 3)/ACN/MeOH	0/2/8–3/12/10–9/65/15–12/2/8–18/2/8	18	1.0
27	Phenyl	H_2_O (pH 3)/ACN/MeOH	0/2/8–3/12/10–9/65/20–12/2/8–18/2/8	18	1.0
28	Phenyl	Ammonium acetate buffer (pH 3.8)/ACN/MeOH	0/2/8–3/12/10–9/65/20–12/2/8–18/2/8	18	1.0
29	Phenyl	Ammonium acetate buffer (pH 3.8)/ACN/MeOH	0/2/8–3/12/10–9/65/20–14/2/8–18/2/8	18	1.0
30	Phenyl	Ammonium acetate buffer (pH 3.8)/ACN/MeOH	0/2/8–3/12/8–9/65/20–16/2/8–18/2/8	18	1.0
31	Phenyl	Ammonium acetate buffer (pH 3.8)/ACN/MeOH	0/2/8–3/12/8–9/65/20–16/2/8–18/2/8	18	1.1
32	Phenyl	Ammonium acetate buffer (pH 3.0)/ACN/MeOH	0/2/8–3/12/8–9/65/20–16/2/8–18/2/8	18	1.1
33	Phenyl	Ammonium acetate buffer (pH 3.0)/ACN/MeOH	0/2/8–3/12/8–9/65/15–16/2/8–18/2/8	18	1.1
34	Phenyl	Ammonium acetate buffer (pH 3.0)/ACN/MeOH	0/2/8–3/12/8–9/65/25–16/2/8–18/2/8	18	1.1
35	Phenyl	Ammonium acetate buffer (pH 3.0)/ACN/MeOH	0/2/8–3/12/8–9/60/20–16/2/8–18/2/8	18	1.1
36	Phenyl	Ammonium acetate buffer (pH 3.0)/ACN/MeOH	0/2/8–5/12/8–9/65/20–16/2/8–18/2/8	18	1.1
37	Phenyl	Ammonium acetate buffer (pH 3.0)/ACN/MeOH	0/2/8–3/12/8–10/65/20–16/2/8–18/2/8	18	1.1
38	Phenyl	Ammonium acetate buffer (pH 3.0)/ACN/MeOH	0/2/8–3/12/8–11/65/20–16/2/8–18/2/8	18	1.1
39	Phenyl	Ammonium acetate buffer (pH 3.0)/ACN/MeOH	0/2/8–4/12/8–9/65/20–16/2/8–18/2/8	18	1.1
40	Phenyl	H_2_O (pH 5)/ACN/MeOH	0/2/8–4/12/8–10/65/20–16/2/8–18/2/8	18	1.1
41	Phenyl	H_2_O (pH 4)/ACN/MeOH	0/2/8–4/20/8–9/65/20–16/2/8–18/2/8	18	1.1
42	Phenyl	H_2_O (pH 4)/ACN/MeOH	0/2/8–9/65/20–16/2/8–18/2/8	18	1.1
43	Phenyl	Potassium dihydrogen phosphate buffer (pH 3.0)/ACN/MeOH	0/5/10–4/20/8–12/65/20–16/5/10–18/2/8	18	1.0
44	Phenyl	Potassium dihydrogen phosphate buffer (pH 5.0)/ACN/MeOH	0/5/10–4/20/8–12/65/20–16/5/10–18/2/8	18	1.0
45	Phenyl	Potassium dihydrogen phosphate buffer (pH 3.0)/ACN/MeOH	0/2/8–4/12/8–12/65/20–16/2/8–18/2/8	18	0.9

C-18 = Octadecylsilane; H_2_O = ultrapure water acidified with anhydrous acetic acid; ACN = acetonitrile; MeOH = methanol.

**Table 2 plants-12-01057-t002:** Best analytical conditions to analyze carajurin by HPLC-DAD-UV.

Equipment	HPLC—Shimadzu Nexera XR^®^
Mobile phase	A: Potassium dihydrogen phosphate buffer (pH 3.0) B: Acetonitrile (ACN) C: Methanol (MeOH)
Column	Ascentis^®^ Phenyl HPLC Column, Supelco (Merck, Darmstadt, Germany)—250 mm × 4.6 mm i.d.; 5 mm particle size
Gradient program	[time(min)/%B/0%C]: 0/2/8–4/12/8–12/65/20–16/2/8–18/2/8
Oven temperature	50 °C
Injection volume	20 μL
Flow rate	0.9 mL/min
Wavelength	480 nm
Carajurin diluent	ACN:MeOH—75:25 (% *v*/*v*)
Retention time (Rt)	14.2 ± 0.2 min
Total analysis time	18 min

**Table 3 plants-12-01057-t003:** Results of intra-day precision analyses of the new analytical method for quantification of carajurin.

Carajurin Standard (µg/mL)	Abs Mean (mAU)	SD ^1^	RSD (%) ^2^
1	66,874.8	162.8	0.2
3	244,951.0	331.9	0.1
6	586,446.3	615.4	0.1

^1^ SD: Standard deviation. ^2^ RSD (%): relative standard deviation. Results are expressed in mAU of absorption at 480 nm. Abs mean = Mean of absorbance (*n* = 3). No significant differences were observed in results (*p* > 0.05) by ANOVA test.

**Table 4 plants-12-01057-t004:** Results of inter-day precision analysis of the new analytical method for quantification of carajurin.

Carajurin Standard (µg/mL)	Day 1			Day 2			Day 3		
	Abs Mean (mAU)	SD ^1^	RSD (%) ^2^	Abs Mean (mAU)	SD ^1^	RSD (%) ^2^	Abs Mean (mAU)	SD ^1^	RSD (%) ^2^
1	66,701.9	880.8	1.3	67,025.2	783.5	1.2	66,897.3	866.2	1.3
3	244,789.8	1598.7	0.7	245,332.8	1287.7	0.5	244,730.6	810.3	0.3
6	585,770.8	3464.6	0.6	586,975.0	2995.3	0.5	586,593.1	2175.9	0.4

^1^ SD: Standard deviation. ^2^ RSD (%): relative standard deviation. Results are expressed in mAU of absorption at 480 nm. Abs mean = Mean of absorbance (*n* = 3). No significant differences were observed in results (*p* > 0.05) by ANOVA test.

**Table 5 plants-12-01057-t005:** Results for accuracy evaluation of the new analytical method for quantification of carajurin.

Theoretical Concentration (µg/mL)	Experimental Abs Mean (mAU) ^1^	Experimental Concentration (µg/mL)	∆ (µg/mL) ^2^	∆ (%) ^3^	∆ Total (%) ^4^
1	56,968	1.10	0.10	9.91	109.91
2	154,157	2.03	0.03	1.34	101.34
3	241,716	2.86	−0.14	−4.59	95.41
4	351,994	3.91	−0.09	−2.13	97.87
5	463,638	4.98	−0.02	−0.39	99.61
6	582,623	6.12	0.12	1.94	101.94

^1^ Experimental mean of absorbance (*n* = 3); ^2^ ∆ (µg/mL) = difference in concentration; ^3^ ∆ (%) = difference in percentage; ^4^ ∆ total (%) = total difference in percentage from theorical concentration. Experimental concentration was calculated from the analytical curve formula described in Linearity.

**Table 6 plants-12-01057-t006:** Robustness evaluation of the new developed analytical method.

	Concentration 1 µg/mL			Concentration 6 µg/mL		
Parameters	Abs Mean (mAU)	SD ^6^	RSD (%) ^7^	Abs Mean (mAU)	SD ^6^	RSD (%) ^7^
Nominal ^1^	67,112	1219	1.82	588,278	2326	0.40
Variation A ^2^	68,156	1607	2.36	589,334	1462	0.25
Variation B ^3^	66,718	467	0.70	584,817	784	0.13
Variation C ^4^	66,739	833	1.25	589,039	2951	0.50
Variation D ^5^	64,849	1155	1.78	588,499	1362	0.23

^1^ Nominal condition: flow rate at 0.9 mL/min; oven temperature at 50 °C; wavelength at 480 nm; acetonitrile brand Tedia (Fairfield, CT, USA). ^2^ Variation A: flow rate at 0.8 mL/min. ^3^ Variation B: oven temperature at 48 °C. ^4^ Variation C: acetonitrile brand Merck (Darmstadt, Germany). ^5^ Variation D: wavelength at 470 nm. ^6^ SD: standard deviation; ^7^ RSD (%): relative standard deviation. Results are expressed in mAU of absorption at 480 nm. Mean of absorbance (*n* = 3).

**Table 7 plants-12-01057-t007:** Results for recovery test of the new analytical method.

Carajurin Standard (µg/mL)	Abs mean Standard Solution (mAU) (RSD %) ^1^	Abs Mean Spiked Solution (mAU) (RSD %) ^1^	Recovery (%)
1	146,840.66 (0.22)	159,426.00 (0.32)	108.57
6	620,157.63 (0.32)	624,071.70 (0.35)	100.63

^1^ RSD (%): relative standard deviation. Results are expressed in mAU of absorption at 480 nm. Mean of absorbance (*n* = 3).

**Table 8 plants-12-01057-t008:** HPLC-DAD methods from the literature to analyze anthocyanidins in *Arrabidaea chica*.

*A. chica* Sample	Mobile Phase and Elution Mode	λmax UV Detector	Column	Flow Rate (mL/min)	References
Crude extract	(A) methanol, (B) methanol: water (1:9) (C) 5% phosphoric acid; Isocratic elution of A:B:C (23:67:10) for 3 min, followed by a linear gradient to A:B:C (90:0:10) in 26 min Total analysis time of 30 min	470 nm	Lichrosorb RP-18 (Merck; 125 mm × 4.1 i.d. mm × 5 µm)	1.2	[6]
Aqueous, ethanolic, and hydroalcoholic (70%) extraction	Analyses were performed according to the methodology described by Devia et al. [6]. Total time analysis of 45 min	470 nm	Phenomenex Gemini C-18 (250 mm × 4.6 mm i.d. × 3 µm)	0.5	[25,26]
Crude extract	A (water + phosphoric acid; pH 2.0 ± 0.10) and B (methanol). Gradient: 0–5 min, 45% B; 5.1–20 min, 90% B; 20–30 min, 100% B From 30 min, 45% B. There is no information about total analysis time.	470 nm	Phenomenex Gemini C-18 (250 mm × 4.6 mm i.d. × 5 µm)	1.0	[39]

**Table 9 plants-12-01057-t009:** 3-Deoxyanthocyanidins (3D-anth) content in *A. chica* leaf extracts (µg/g dried plant material) expressed as carajurin.

3D-Anth	Rt (Min)	Area (mAU)	Concentration (µg/g)
(1)	11.8	1,501,380 ± 6342.53	148.85 ± 0.60
(2)	12.5	2,415,521 ± 13,128.54	236.10 ± 1.25
(3) ^1^	14.2	7,544,136 ± 12,282.95	725.61 ± 1.17
(4)	16.2	3,031,838 ± 23,327.25	294.93 ± 2.22

^1^ (3) = carajurin. Values are expressed as the mean ± SD (*n* = 3, see experimental); Rt = retention time; 3D-anth 1–4: see Figure 3B.

## Data Availability

Not applicable.

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
