# Peer review of "Validation of a New HPLC-DAD Method to Quantify 3-Deoxyanthocyanidins Based on Carajurin in Medicinal Plants and for Chemical Ecology Studies"

_plants, 2023, doi:10.3390/plants12051057_

Round 1
Reviewer 1 Report
The authors developed and validated a new analytical HPLC method for the quantification of quantify 3-deoxyanthocyanidins based on the carajurin content in plant matrices of some medicinal plants. The method proved to be reliable and reproducible.
Title:
It is recommended to reduce the title to less than 20 words.
Abstract:
The topic is well contextualized, the objective of the study is clear, the methods are described, some results are presented, and some conclusions are presented.
1) Line 21. writing in the third person
Keywords:
OK. It is recommended to increase the number of keywords (it could improve the visualization of the manuscript). The magazine suggests between 3 and 10 words
Introduction:
1) Line 54: Writing in the third person
2) A very good contextualization is presented, the authors support the section with bibliographic references of scientific quality. The knowledge gap is well defined and shows the contribution of the research to the development of new knowledge.
Results and discussion:
The results are clear and convincing, the experimental quality of the data guarantees a good analysis and allows evidence of the quality of the work presented.
1) lines 113, 120. writing in the third person
2) line 175. cite the references adopted
Materials and Methods:
The materials used are well described, the methods are clear and can be used in other laboratories. The described methodology allows to demonstrate the experimental quality of the reported data.
Conclusions:
Writing in the third person
Reviewer 2 Report
The present manuscript entitled "Development and validation of a new HPLC-DAD method based on carajurin content to quantify 3-deoxyanthocyanidins in medicinal plants and for chemical ecology studies" by Paulo Victor Ramos de Souza, Vanessa Gonzalez Ferreira Caminha Martins, Maria do Socorro dos Santos Chagas, Carla Junqueira Moragas-Tellis, Maria Dutra Behrens, and Davyson de Lima Moreira (plants-2223600) is written correctly and has a good structure; moreover, it has all the necessary parts. The article is interesting from an analytical and environmental point of view; therefore, it should interest the reader. I proposed improvements in the method description and with a presentation of figures. The paper meets Plants' requirements, and I recommend the article for publication in Plants following the common editing stage. My current decision is a minor revision. More specific comments and observations are presented below.
1. Why is the first author highlighted in color?
2. Page 1, lines 28 and 29. You can specify values for specific parameters.
3. On what basis was the wavelength of 480 nm chosen? Have other wavelengths been considered?
4. Were experimental design methods used in the optimization?
5. Table 1. Abbreviations that have not been entered before appear.
6. Figures with chromatograms. A description of the axis should be added along with the units, and redundant text should be removed. You can add a description to the peak from the analyte.
7. Page 6, line 160. What can be done in the event of strong interference effects? How would you deal with them? What types of interference effects could occur?
8. Figures 3 and 4 can be merged and presented in a single figure.
9. RSD expressed as a percentage is the coefficient of variation (CV). This is currently mixed up in text and tables.
10. Page 11, line 314. Liters are written as "l"; it should be "L". Units must be unified.
11. The "relationship" is mentioned. This term should be changed to "relation". The relationship tends to be used more broadly to describe the interactions between specific people or smaller groups of people.
12. Page 12, line 324. What were the parameters of the water used?
13. Does the conducted studies have disadvantages?
14. Conclusions. Please clearly highlight the most important advantage.
15. Appropriate tools should be used to best characterize the method when developing a new approach (e.g., AGREE- Analytical GREEnness Metric Approach or RGB model)
16. References. [1] is empty. Everything was changing.
I hope that the comments presented will help improve the article.
